# Conciliator steering: Imposing user preference in multi-objective reinforcement learning

**Sara Pyykölä**                                                   *sara.pyykola@helsinki.fi*

*Department of Computer Science*
University of Helsinki, Finland

**Klavdiya Bochenina**                                            *klavdiia.bochenina@helsinki.fi*
*Department of Computer Science*
University of Helsinki, Finland

**Laura Ruotsalainen**                                            *laura.ruotsalainen@helsinki.fi*
*Department of Computer Science*
University of Helsinki, Finland

**Reviewed on OpenReview:** `https://openreview.net/forum?id=XAD2kcBS50`

## Abstract

Many real-world problems with multiple objectives require reinforcement learning solutions that can handle trade-offs in a user-preferred manner. In the multi-objective framework, a single algorithm adapting to different user preferences based on a pre-defined reward function and a subjectively defined scalarisation function may be developed. The scalarisation function approximation can be done by fitting a meta-model with information gained from the interaction between the user and the environment or the agent. The interaction requires exact formulation of a constructive feedback, which is also simple for the user to give. In this paper, we propose a novel algorithm, Conciliator steering, that leverages priority weighting and reward transfer to seek optimal user-preferred policies in multi-objective reinforcement learning under expected scalarised returns criterion. We test Conciliator steering on DeepSeaTreasure v1 benchmark problem and demonstrate that it can find user-preferred policies with effortless and simple user-agent interaction and negligible bias, which has not been possible before. Additionally, we show that on average Conciliator steering results in a fraction of carbon dioxide emissions and total energy consumption when compared to a training of fully connected MNIST classifier, both run on a personal laptop.

## 1 Introduction

Multi-objective reinforcement learning (MORL) problems have been gathering more and more attention, as shown by the extensive survey of Hayes et al. (2022a). Multi-objective problems appear naturally in real-world problems, where many goals need to be achieved at once instead of striving for only one. In team projects, the individual can increase team's benefit by sacrificing their own benefit and spending more effort. While driving in a congested traffic, the driver must maintain safety and still optimize the travel time. In games, the player needs to weigh different strategies and their risks and gains. Multi-objective reinforcement learning is an excellent approach to such problems given its suitability to simulations instead of data-driven methods. Yet it has generally not been utilized to its full extent, as noted by Hu et al. (2023). When used, the multiple objectives are often reduced to single objective to lessen computational costs, leading to a single optimal way to proceed when in practice, there are many different ways to proceed, each with unique pros and cons. As such, the optimal way is inherently subjective, varying across the workers, drivers and players' preferences, but the solutions offered by MORL to each individual may not vary.

When encountering the need for tailor-made propositions in decision-making, MORL has primarily offered three methods: a priori information about individual and contextual preferences; secondly, a manual empirical testing to produce the preferred outcome for each case; or thirdly, interactive information retrieval about preferences. For the worker for example, these approaches could correspond to having an order of tasks defined by the supervisor, taking a guess which task to tackle first or discussing with the teammates and supervisor what tasks could be completed first. The last method offers most flexibility without increasing computational costs significantly, unlike the second method with highest costs and the first method with the least flexibility. The difficulty of the last method, however, lies in facilitating the interaction in a way that benefits the MORL solution instead of hindering it with bias, as shown by Bradley Knox & Stone (2008).

Another problem arises from the complexity of context and the end purposes of the MORL solution. The team and their workers may connect only remotely, necessitating different work practices than a team meeting face-to-face only. The team may have to undergo the same discussion weekly while meeting monthly milestones, thus giving an option to catch up for the previous week's stagnation in the upcoming weeks instead of making a steady progress. The MORL solution would have to adapt to this context and repeating use, which is not straightforward to perform. Interactive information retrieval about preferences of the user of MORL algorithm holds potential to solve this aspect, but only with the cost of bias to the MORL solution.

In this paper, the aforementioned challenges of effortless and unbiased human preference elicitation in a computationally inexpensive manner shall be addressed by proposing a novel steering algorithm named Conciliator steering. It takes advantage of priority order and reward transfer to approximate the user's subjective preferences and produce user-preferred policies based on this information. We perform an experimental study of Conciliator steering in the DeepSeaTreasure v1 benchmark by Cassimon et al. (2022), and note that Conciliator steering produces satisfactory policies, while being simple for the user to interact with. The Conciliator steering is also computationally lightweight: on average, it produces only a promille of carbon dioxide emissions and two promilles of total energy consumption compared to a training of a fully-connected MNIST classifier Bouza et al. (2023), when the CodeCarbon library developed by Courty & Schmidt (2023) is used for the estimation and a personal computer is used as the environment.

This paper is divided into four sections. First, the related work is presented in Section 2 and a problem formulation is given in Section 3. Afterwards, the proposed solution is introduced in Section 4, an experiment is presented in Section 5 and results reported in Section 6. Finally, a discussion about its characteristics is given in Section 7, and conclusions and steps for future research are suggested in Section 8.

## 2 Related work

### 2.1 Multi-objective reinforcement learning in decision-making problems

The use of RL in multi-objective decision-making has been explored in numerous studies recently, as noted by Hayes et al. (2022a) and Roijers et al. (2013) in their surveys. Among other characteristics, the proposed solutions can be categorised according to the number of policies found: single-policy or multi-policy, see e.g. Hu et al. (2023). Single-policy methods use a linear scalarisation function to reduce the multi-objective Markov decision process (MOMDP) into a single-objective Markov decision process with only one optimal solution. Multi-policy methods in turn formulate various scalarisation functions, resulting in a set of optimal policies approximating the whole Pareto front. While single-policy methods have demonstrated their prowess in various tasks after the milestone study by Mnih et al. (2013), they suffer from manual and laborious testing to find suitable weights for the scalarisation function. Additionally, as proven by Vamplew et al. (2008), applying a linear scalarisation function can limit the found solution set.

---

**Definition 1** (Pareto optimality)**.** *The reward vector $\mathbf{r} \in \mathbb{R}^n$ is Pareto optimal, if no reward vector component $\mathbf{r}_i$ can be increased without decreasing another reward vector component $\mathbf{r}_j, j \neq i, i, j \in \{0, 1, ..., n\}$.*

---

Using a MOMDP framework without a linear scalarisation function holds potential especially for conflicting objectives the existing MORL struggle to optimize, as these objectives can not be Pareto optimal at the same

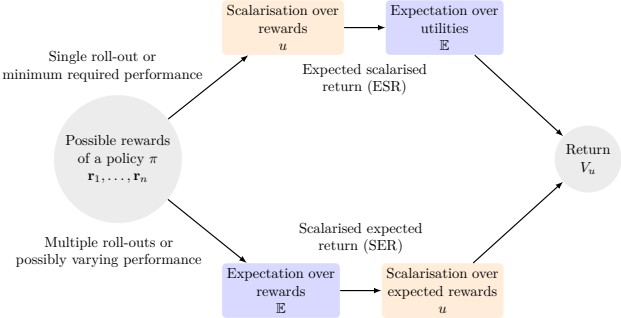

Figure 1: A diagram of the two different optimality criteria used in MORL. The chosen criterion depends on the purpose and use of the optimal policy. The order of computations differs depending on the chosen criterion, making it crucial to have well-defined intermediate calculations for the accurate computation of return.

time according to Definition 1, and thus require a trade-off to be made Hu et al. (2023). The formulation of suitable trade-offs is further complicated by the different scales of rewards present in the reward function, giving the agent the possibility of formulating policies accumulating only large reward components while ignoring the small reward components in the reward vector. The resulting policy in turn doesn't represent the optimal decision for the human using the algorithm. The primary solution presented so far in the literature is often referred to as reward shaping or reward engineering, i.e. the careful hand-crafting of the reward and scalarisation functions to match the scales of returns to the human's preference, as noted by Hu et al. (2023) and Hayes et al. (2022a). These hand-crafted approaches suffer from the bane of the single-policy methods: extensive empirical testing is required to make them work. While allowing multiple objectives to exist with an objective reward function, alternative scalarisation functions could potentially be learned in the optimization, eliminating the need for reward shaping and engineering. Thus there is a need for new MORL methods that address these issues of laborious empirical testing, conflicting objectives in the optimization and different possible scalarisation functions when using the multi-objective reinforcement learning framework.

## 2.2 Expected scalarised returns and scalarised expected returns

To formulate an optimal policy in a MOMDP framework, criteria for optimality must first be given. Hayes et al. (2022a), in accordance to the existing studies across different disciplines, categorised the optimality criterion into two classes according to their mathematical definitions and practical end uses: expected scalarised returns (ESR) and scalarised expected returns (SER). In ESR, the optimality is induced by one roll-out of the optimal policy instead of multiple roll-outs, and the returns are calculated from the rewards by first applying the scalarisation function and then the expectation. In SER, the optimality is induced from many roll-outs and the order of operations is reversed, as shown in Figure 1. This choice affects the solution set, as illustrated in Table 1, where an user with a second-degree scalarisation function must choose from two options, each having two different outcome pairs with specific rewards: having an ESR criterion leads to a different optimal policy with respect to the SER criterion. In practice, the conditions also induce different returns in roll-outs: in ESR, the returns have a strict minimum performance, while in SER, the returns have a good average performance.

When assuming the ESR criterion, the topic of unknown scalarisation function remains little studied in RL. A study by Hayes et al. (2022b) applied first-order stochastic dominance into a RL framework, proving that the component-wise highest reward vector has the highest expected return under a monotonically increasing scalarisation function. This result admittedly can be applied in many places, but in the other hand neglects the source of utility information available from the user in less ambiguous situations, thus wasting an opportunity to formulate the user preferences more computationally efficiently. The significance of this information is illustrated in Table 1, where two different non-linear utility functions lead to different optimal policies for the user when operating under the ESR criterion. A second study by Hayes et al. (2021) used distributional Monte Carlo tree search algorithm to solve the multi-objective Markov decision process

Table 1: Tables depicting a practical example of returns under the two different optimality criteria in the presence of two different non-linear scalarisation functions, where the available action is to choose a game with two different outcomes and rewards out of two different games. The optimal policies are underlined, showing that the user has a different optimal policy depending on the used optimality criteria and scalarisation function.

(a) Probabilities and rewards for outcomes $A$ and $B$ in each game.

|  | $\mathbf{r}_A$ | $P(A)$ | $\mathbf{r}_B$ | $P(B)$ |
|---|---|---|---|---|
| Action 1 | $(3,5)$ | $0.5$ | $(3,2)$ | $0.5$ |
| Action 2 | $(2,9)$ | $0.2$ | $(3,1)$ | $0.8$ |

(b) The user's return calculation with equal priorities in the scalarisation function.

Scalarisation function $u\colon \mathbb{R}^2 \to \mathbb{R}, u(x_1, x_2) = x_1^2 + x_2^2$

|  | ESR | | SER | |
|---|---|---|---|---|
|  | $u(\mathbf{r}_A), u(\mathbf{r}_B)$ | $\mathbb{E}(u(\mathbf{r}))$ | $\mathbb{E}(\mathbf{r})$ | $u(\mathbb{E}(\mathbf{r}))$ |
| Action 1 | $(34, 13)$ | $23.5$ | $(3, 3.5)$ | $\underline{21.25}$ |
| Action 2 | $(85, 10)$ | $\underline{25}$ | $(1.2, 2.6)$ | $8.2$ |

(c) The user's return calculation with unequal priorities in the scalarisation function.

Scalarisation function $u\colon \mathbb{R}^2 \to \mathbb{R}, u(x_1, x_2) = 0.5x_1^2 + 5x_2^2$

|  | ESR | | SER | |
|---|---|---|---|---|
|  | $u(\mathbf{r}_A), u(\mathbf{r}_B)$ | $\mathbb{E}(u(\mathbf{r}))$ | $\mathbb{E}(\mathbf{r})$ | $u(\mathbb{E}(\mathbf{r}))$ |
| Action 1 | $(17, 65)$ | $\underline{41}$ | $(3, 3.5)$ | $33.5$ |
| Action 2 | $(42.5, 50)$ | $27.4$ | $(1.2, 2.6)$ | $\underline{34.52}$ |

under ESR criterion, but in this implementation, the explicit scalarisation function is required to be known a priori in comparison. While this avoids the issue of computational burden for the formulation of scalarisation function, there is little to no guarantee that the a priori definition is a match for the user's scalarisation function. To ensure this condition is met, more interaction between the human and agent is required, which leads to human-centered RL.

## 2.3 User preference in multi-objective reinforcement learning

Continuing with idea of the user preference with respect to the optimal solution, the field of interactive RL must be raised to light. This field utilises human feedback to elicit preferences that can be then used to guide the RL agent in different ways. The mathematical concept of eliciting human preferences to formulate an optimal solution in multi-objective decision-making has been long around, as can be noted from the paper of Wierzbicki (1982) presenting a rigorous mathematical framework bringing many works together at the time. Different approaches how to integrate human knowledge into RL in game applications has been surveyed by Sutton & Barto (2018) as a part of their book. A more detailed survey of human-centered RL only has been written by Li et al. (2019).

However, the empirical testing of human-centered RL has emerged fairly recently. A milestone in this regard was conducted by Mannor & Shimkin (2004), when they tested the idea of geometric steering for scalar rewards in stochastic games. A notable modern study is the Q-steering algorithm by Vamplew et al. (2017), where the user specifies their preference (target or reference point) in the bi-objective reward space and the algorithm determines a mixture of non-stationary policies converging on that reward. While the two aforementioned empirical studies present a viable approach for the user feedback, they are hindered by their assumptions: Mannor and Shimkin do not consider multi-objective problems, whereas Vamplew et al. require the base policies along with their returns and the final outcome is a linear mixture of those returns, which

ignores the non-linear mixtures altogether. The logical next question about whether the user's reward signal is an absolute point for the RL agent to converge to, or only a guiding factor, is a very reasonable question, as pointed out by Li et al. (2019) in their survey and MacGlashan et al. (2023) in their study regarding the bias of human feedback. Further complicating the matter is the definition of Pareto optimality, as the user can compensate for one reward component by decreasing others, and if no validation is made, the selected policy might be sub-optimal with respect to the Pareto optimality. It is important that this question is thoroughly addressed in all human-centered RL methods, and more research into the topic would be beneficial for the advancement of RL.

Other examples of RL utilising human feedback include Wanigasekara et al. (2019), who use personalized search rankings to elicit user preferences; Roijers et al. (2017), who use interactive Thompson sampling and user-environment interactions; and Roijers et al. (2021), who use Gaussian process utility Thompson sampling for continuous cases. Ikenaga & Arai (2018) in contrast utilise inverse RL, while Saisubramanian et al. (2020) use random queries, approval, corrections, and demonstrations to formulate the user preferences. Each of these approaches is cumbersome or practically non-trivial for the user to carry out, and a more straightforward and flexible interaction with the environment would pose a remarkable enhancement in terms of human effort and computation.

Several studies of human-centered RL have used the rewards as an interface. Thomaz & Breazeal (2008) explicitly investigated whether the human-induced reward is suitable for the RL agent to use in a robotics application and gave a several measures to improve the agent's performance while using human-induced rewards. Bradley Knox & Stone (2008) in turn use scalar reward signals from the human to play Tetris. Loftin et al. (2016) present a probabilistic model of human's rewards and punishment, and apply two learning algorithms over it in order to learn even from the absence of human's reward. However, the explicit usage of human-induced scalar reward is non-trivial, as they can transmit bias for the RL agent, harming the convergence to the optimal policy, as noted by MacGlashan et al. (2023). They employ an actor-critic model to remove the bias in human's reward feedback caused by the agent's current policy. Thus concluding, it is more viable to use a more comprehensive and protected interface that prevents the bias.

Another research avenue of human-centered solutions has regarded altruistic MORL algorithms. The first study to note is the Lorenz optimality by Perny et al. Perny et al. (2013), where a mathematical foundation and methods to approximate Lorenz dominant solutions are proven. However, this framework axiomatically assumes that the most optimal and fair policies produce most uniformly distributed rewards, which may not be true for all users in problems with complex trade-offs present. Another study about altruistic MORL has been conducted Franzmeyer et al. Franzmeyer et al. (2022), but their assumption is that the altruism extends the action space, which is not applicable to fully observable state and action spaces, where the latter is static.

The suitable and optimal approach to human alignment in multi-objective decision-making in RL is not the only hurdle to cross however. Often difficulties arise from in the case of longer trajectories with sparse rewards Moerland et al. (2023), where planning over the policy is required from the agent to find optimal policies. This research question is addressed by model-based reinforcement learning.

## 2.4 Model-based reinforcement learning

Model-based reinforcement learning aims to define a model, which offers reversible access to environment dynamics: the possibility to estimate an action's consequences in the environment Moerland et al. (2023). The core motivation for the environment model lies in the policy planning: by having knowledge of future states, the agent can plan its policy for longer time intervals than two consecutive states. To better enable this, studies by Ha & Schmidhuber (2018), have separated the environment model and the policy selection into their own separate modules in the solution architecture. This design choice has produced promising results in the studies, implying that good and critical policy selection and accurate and useful environment model are equally important in achieving optimal long-term policies.

This reasoning then raises the question what gives rise to an accurate and useful environment model. In recent studies, deep learning has showed significant potential. Recurrent neural networks have been utilised by Chiappa et al. (2017) and Ha & Schmidhuber (2018), while many other models have been compared by Kaiser et al. (2019). Recent progress in the field has been made by Google Deepmind's Adaptive Agents

Adaptive Agent Team (2023), which utilised a transformer architecture to predict the environment for four next steps. That said, Monte Carlo techniques have also been used with success in multi-objective problems by Coulom (2007) and Hayes et al. (2021). Notable cons of using environment models are the accuracy of the environment model and the optimization of the model and its possible training, where the former can be difficult to define and the latter can increase the computational cost from the already high cost of RL.

## 2.5 Summary

Summarizing, it can be concluded that there is a definite gap in MORL research regarding decision-making problems: a multi-policy method operating under ESR condition that can adapt to different scalarisation functions. The scalarisation function can be interpreted as the user's preference, but an effortless and simple way to exact this preference does not yet exist or it may cause bias for the agent. Various assumptions can help to solve the problem, but they limit the problem formulation unreasonably, reducing the methods' applicability. Our research is designed to bridge this gap, and incite new research in human-centered MORL research utilising model-based RL, whose results can be used in real-world applications. To begin this, we will present a MOMDP framework as our problem formulation, and then present our proposed solution, Conciliator steering, named in reference to its ability to deal with conflicting objectives while building on ideas presented in the Q-steering by Vamplew et al. (2017), reference point technique by Wierzbicki (1982) and the Lorenz set of Perny et al. (2013).

## 3 Problem formulation

The multi-objective Markov decision process is represented by the tuple $\langle S, A, T, \mu, \gamma, \mathbf{R} \rangle$ where $S$ is the state space, $A$ is the discrete action space, $T \colon S \times A \times S \to [0,1]$ is a probabilistic transition function from a state $s$ to a state $s'$ when an action $a$ is taken. The transition can be either deterministic, when the probability is always one for one specific state and zero for others, or stochastic, resulting in a probability distribution of possible states to be transitioned into with a probability $p$. The $\gamma \in [0,1)$ is a discount factor for future rewards, $\mu \colon S \to [0,1]$ is a probability distribution over initial states, and $\mathbf{R} \colon S \times A \times S \to \mathbb{R}^n$ is a reward function, specifying the reward vector $\mathbf{r}$ for a given action $a$ in the starting state $s$ and ending state $s'$, with one component for each of the $n$ objectives Hayes et al. (2022a).

Now it is defined that the agents act according to a policy $\pi \in \Pi$, where $\Pi$ is the set of all possible policies. A policy $\pi$ is a mapping $\pi \colon S \times A \to [0,1]$, and the value function of the policy $\pi$ is then the following,

$$V^\pi = \mathbb{E}\left[\sum_{k=0}^{\infty} \gamma^k \mathbf{r}_{k+1} | \pi, \mu\right],$$

where $\mathbf{r}_{k+1} = \mathbf{R}(s_k, a_k, s_{k+1})$ is the immediate reward received at time step $k+1$. Additionally, a scalarisation function $u \colon \mathbb{R}^d \to \mathbb{R}$, that maps a multi-dimensional reward vector $\mathbf{r}$ to a single scalar value, is defined. This problem is then solved by finding an optimal user-preferred policy $\pi^*$ that maximises the return for an agent. As an optimality criterion in a MORL case, two alternatives can be used: the expected scalarised returns (ESR),

$$V_u^\pi = \mathbb{E}\left[u\left(\sum_{k=0}^{\infty} \gamma^k \mathbf{r}_k\right) | \pi, s_0\right],$$

or the scalarised expected returns (SER),

$$V_u^\pi = u\left[\mathbb{E}\left(\sum_{k=0}^{\infty} \gamma^k \mathbf{r}_k\right) | \pi, s_0\right].$$

An example how each criterion is calculated is shown in Table 1, detailing the input of scalarisation function and expectation closely. The ESR criterion is applied when the optimal policy is executed only once and the SER is applied when the optimal policy is executed multiple times. This difference is illustrated in Figure 1. The sets of optimal policies for these criteria are also not equivalent when considering non-linear scalarisation functions $u$, as shown in Table 1. Consequently different policies should be utilised for each criterion.

It should be noted here that the mathematical formulation for the set of optimal policies is an open question under ESR. In this paper, due to the assumption of user preference dictating the optimal policies, this question will remain open.

# 4  Proposed solution

Let's assume we have a decision-making problem cast as a MOMDP framework with the ESR criterion as presented in Section 3. Then the optimal policy $\pi^*$, the solution to the problem, should produce results that meet the decision-maker's goals for the outcome. The very first hurdle in a such problem would be to understand the environment at play; what variables affect the problem and how, so we could predict how each action affects the desired outcome and plan the optimal policy for long time intervals. The second hurdle is then using this information to select the optimal user-preferred policy.

The first step, the estimation of the environment model, is left for the Approximator, which is the first part of the proposed solution's architecture. This kind of task is the primary problem of model-based RL Moerland et al. (2023), and as such, any findings of the field can be utilised to solve the task. A formal definition for the Approximator is given below.

---

**Definition 2** (Approximator). *An Approximator is a tool that estimates the episodic reward for each policy in a given state.*

---

The second step, the policy selection part, is left for Conciliator, the second part of the proposed solution's architecture. This task falls on the fields of multi-objective decision-making and user preferences in RL, so findings from these fields can be utilised to solve the task. A formal definition for the Conciliator is given below.

---

**Definition 3** (Conciliator). *A Conciliator is a tool that searches for the user's preferred episodic reward $\mathbf{r}'$ out of all episodic rewards $\mathbf{r}$.*

---

Our proposed solution for the user-preferred policy selection uses rewards to incorporate the user preference into the scalarisation function approximation. As can be seen from the works of Thomaz & Breazeal (2008) and Bradley Knox & Stone (2008), the integration of human feedback into the design of the reward function in a controlled way can lead to more flexible and efficient exploration of different reward functions. Consequently, the approximation of scalarisation function presents a dual opportunity: the scalarisation function information can be leveraged both for defining the user preference across different tasks' reward functions and for different scalarisation functions across individual users in the same task. While the generality and efficiency of different environment models is an active research question as noted by Moerland et al. (2023), there is a theoretical possibility to re-use the environment models as well, resulting in a easily transferable RL pipeline that accounts for user preferences in decision-making problems.

To formalize how Conciliator locates the user-preferred episodic reward $\mathbf{r}$ and how it is used in the approximation of the scalarisation function, we first define a transfer vector $\mathbf{t}$, a priority weighting $\mathbf{p}$ and a user profile.

---

**Definition 4** (Priority weighting). *A priority weighting is a non-negative real vector that signals the objectives' importance from the user's viewpoint:*

$$\mathbf{p} = (a_1, a_2, \ldots, a_n).$$

---

> **Definition 5** (Transfer vector). *A transfer vector $\mathbf{t} \in \mathbb{R}^n$ is the difference between an episodic reward vector $\mathbf{r}$ and the user's preferred episodic reward vector $\mathbf{r}'$:*
>
> $$\mathbf{r}' = \mathbf{r} + \mathbf{t} \Leftrightarrow \mathbf{t} = \mathbf{r}' - \mathbf{r}.$$

> **Definition 6** (User profile). *A user profile is the set of user's priority weighting $\mathbf{p}$ and the corresponding preferred episodic reward $\mathbf{r}'$, given the episodic reward $\mathbf{r}$.*

When the user specifies a priority weighting over the episodic reward $\mathbf{r}$ used as a baseline, the transfer $\mathbf{t}$ is optimized so that the lower priority objectives will transfer their reward to the higher priority objectives and vice versa. After the transfer $\mathbf{t}$ is calculated, the preferred episodic reward $\mathbf{r}'$ can be calculated given the $\mathbf{t}$ and $\mathbf{r}$. Thus, by testing different priorities and consequently different transfers depending on the priorities, the user-preferred episodic reward $\mathbf{r}'$ can be found as an end result.

Put formally, given the $\mathbf{r}$ and $\mathbf{p}$, we minimize the transfer $\mathbf{t}$ from the following equation under two constraints, where $\varepsilon$ is an arbitrary machine epsilon and $\bar{\mathbf{r}}$ is the mean of the reward vector $\mathbf{r}$:

$$\sum_{i=1}^{n} \left( \mathbf{r}_i + \mathbf{t}_i - \bar{\mathbf{r}} \cdot \mathbf{p}_i \right)^2 \text{s.t.} \tag{1}$$

$$\sum_{i}^{n} \mathbf{p}_i = n, \forall \mathbf{r}_i : \mathbf{r}_i \geq 0, \exists i : \mathbf{r}_i > 0 \text{ and } \left| \sum_{i}^{n} \mathbf{t}_i \right| \leq \varepsilon. \tag{2}$$

Here we should note the special case of equal priorities between the objectives, which leads Equation 1 to being a variance minimization problem. The solution set is then proven to be a Lorenz set by Perny et al. (2013).

Additionally, there are two special constraints of the transfer vector to note. Firstly, in its current formulation, it only works for positive rewards. In the case of negative rewards, the optimization would have to be re-formulated component-wise and the definition of this optimization is not straightforward, as the distance function used in the optimization would have to account for it. The negative rewards can be transformed to similar positive ones to circumvent this aspect. Secondly, the scale of transfer between the rewards is currently one-to-one, necessitating the scale of the rewards be roughly similar to one other so that one reward can not gain infinity by so that one reward component cannot gain infinity by transferring only a small reward from other reward component. This too can be achieved by transforming the rewards before the optimization. It is essential to choose bijective transformations however, so the transformations can be effortlessly inverted later on when the preferred episodic reward is given back to the user.

However, this user-preferred reward $\mathbf{r}'$ can be guaranteed to be an attainable target point for optimization in the reward space, reducing the resulting bias to the RL solution's convergence considerably. This is mainly due to the linear constraint $\sum_{i=1}^{n} \mathbf{p}_i = n$ posed in Equation 2. With the constraint, the transfer is taken only from an existing episodic reward $\mathbf{r}$. This can be proven to result in possible preferred episodic rewards $\mathbf{r}'$ that are bounded in distance with respect to the baseline reward $\mathbf{r}$ as follows. Let $\mathbf{p} = n \cdot \mathbf{e_k}, \mathbf{t} = \left( \sum_{i=1}^{n} r_i + r_k \right) \mathbf{e_k} - \mathbf{r}$ and $\mathbf{r}' = \sum_{i=1}^{n} r_i \cdot \mathbf{e_k}$, where $\mathbf{e_k} \in \mathbb{R}^n$ is the $n$-dimensional unit vector for the $k$th reward vector component $r_k$, to where all the reward from other reward components $r_j, j \neq k$, is being transferred from. Then the

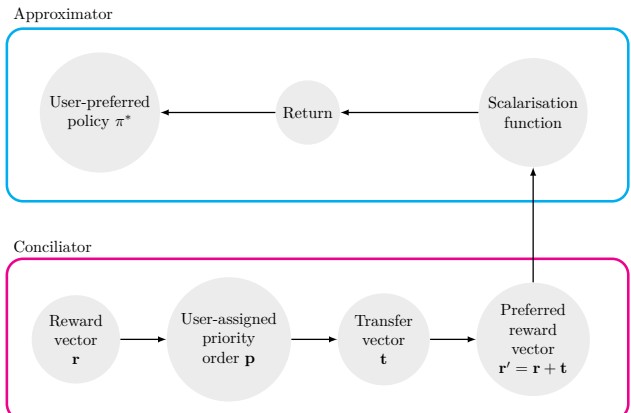

Figure 2: A flow diagram of the proposed solution. The parts of the proposed solution are outlined with blue and pink and the mathematical definitions of the proposed solution are marked with gray circles. In the proposed solution, the user feedback is used in the calculation of the scalarisation function, which then determines the user-preferred policy.

distance between $\mathbf{r}$ and $\mathbf{r}'$, denoted by $d(\mathbf{r}, \mathbf{r}')^2$, is the following:

$$
\begin{aligned}
d(\mathbf{r}, \mathbf{r}')^2 = d(\mathbf{r}, \mathbf{r} + \mathbf{t})^2 &= \|\mathbf{t}\|_2^2 \\
&= \|\left(\sum_{i=1}^n r_i + r_k\right) \mathbf{e_k} - \mathbf{r}\|_2^2 \\
&= \|\left(\sum_{i=1}^n r_i + r_k\right) \mathbf{e_k}\|_2^2 + \|\mathbf{r}\|_2^2 - 2\left(\sum_{i=1}^n r_i + r_k\right) \cdot \mathbf{r} \cdot \mathbf{e_k} \\
&= \left(\sum_{i=1}^n r_i + r_k\right)^2 + \|\mathbf{r}\|_2^2 - 2r_k \left(\sum_{i=1}^n r_i + r_k\right) \\
&= \left(\sum_{i=1}^n r_i\right)^2 + (r_k)^2 + 2r_k \sum_{i=1}^n r_i + \|\mathbf{r}\|_2^2 - 2r_k \sum_{i=1}^n r_i - 2(r_k)^2 \\
&= (n\bar{r})^2 + \|\mathbf{r}\|_2^2 - (r_k)^2
\end{aligned}
$$

The last equation attains the maximum when $r_k = \min_i r_i$. $\square$

Secondly, in order to solve the problem posed by Equation 1 under the constraints defined in Equation 2, it can be proven that the transfer vector has a global minimum from the positive definite Hessian of the transfer vector:

$$
H(\mathbf{t}) = \begin{bmatrix} \frac{\partial^2 \mathbf{t}}{\partial t_1^2} & \cdots & \frac{\partial^2 \mathbf{t}}{\partial t_1 \partial t_n} \\ \vdots & \ddots & \vdots \\ \frac{\partial^2 \mathbf{t}}{\partial t_n \partial t_1} & \cdots & \frac{\partial^2 \mathbf{t}}{\partial t_n^2} \end{bmatrix} = \begin{bmatrix} \frac{\partial 2\mathbf{e_1 t}}{\partial t_1} & \cdots & \frac{\partial 2\mathbf{e_1 t}}{\partial t_n} \\ \vdots & \ddots & \vdots \\ \frac{\partial 2\mathbf{e_n t}}{\partial t_1} & \cdots & \frac{\partial 2\mathbf{e_n t}}{\partial t_n} \end{bmatrix} = \begin{bmatrix} 2 & \cdots & 0 \\ \vdots & \ddots & \vdots \\ 0 & \cdots & 2 \end{bmatrix} = 2\,\mathbf{I},
$$

where $\mathbf{I}$ is the identity matrix and $\mathbf{e_k} \in \mathbb{R}^n$ the $n$-dimensional unit vector for the $k$th vector component. This ensures that the transfer vector transforming the episodic reward $\mathbf{r}$ to the user's preferred episodic reward $\mathbf{r}'$ exists and is unique.

Thus to find the policies closer to the user-preferred episodic reward $\mathbf{r}'$, Equation 1 can be minimized globally using a suitable optimization algorithm or by calculating the gradient of the transfer vector, which will provide the transfer vector required. For our case, we have chosen the simplicial homology algorithm for

---

**Algorithm 1** Conciliator steering

---

**Require: r**

    The user assigns the priority order **p**

    The transfer vector **t** is computed using SHGO

    The preferred reward vector **r′** is computed

    Choose a distance function between **r** and **r′** as the scalarisation function

    Use the priority order as weights for the scalarisation function

    Search the optimal policies using the scalarisation function, calculating the returns

---

Lipschitz optimisation (SHGO) introduced by Endres et al. (2018) as our optimization algorithm for several reasons. Firstly, the problem formulation posed in Equations 1 and 2 meets the conditions of the SHGO; secondly, SHGO is able to solve the issue of a good initial value or black-box optimization for a possibly non-convex solution; finally, it is capable of handling 10 objectives at a time while being fast in run time.

Now that the user's preferred episodic reward $\mathbf{r}' = \mathbf{r} + \mathbf{t}$ and additionally a priority weighting **p** over objectives is known, only the question of finding the optimal policy using this information stands to be solved. Hayes et al. (2022a) claim that in cases where there is uncertainty about the user's scalarisation function, the RL practitioner should refrain from making an exact formulation of the scalarisation function. Here we would like to make a counter-argument: no matter the exact definition of the user's scalarisation function, it is likely to be a some kind of distance function to the preferred episodic reward $\mathbf{r}'$, increasing utility when the reward received in the time step $t$ starts to close in on the preferred episodic reward. While the user indeed may not be able to elaborate the specific formulation of the distance function, it is relatively straightforward for the RL practitioner to define a distance function as a viable scalarisation function, as suggested by Wierzbicki (1982). The distance functions also fulfill the criteria of monotonicity of utility with respect to rewards: the utility increases when the reward is closer to the user's preferred reward, which is intuitive for optimality in human-centered RL. The difficulty in this approach lies in the bias of the preferred episodic reward: should the user ask for a somehow sub-optimal episodic reward, they can receive a policy producing resulting in that episodic reward exactly regardless of the implicit sub-optimality. However, various ways to account for the optimality can be studied — Vamplew et al. (2017) for example choose the user's preferred rewards so that they correspond to the same average return as the resulting rewards from Pareto optimal policies — and the exact formulation of scalarisation function is clearly the most straightforward and computationally efficient approach to solve an MOMDP in a user-preferred manner. Thus we propose the scalarisation should be performed in general fashion, by choosing the distance function between the episodic reward **r** and the user's preferred episodic reward $\mathbf{r}'$ as the scalarisation function, as suggested by Wierzbicki (1982).

In our method, we have used a reciprocal of $l_1$-norm with the normalized priority weighting as the distance function, leading to a scalarisation function that increases fastest at the direction of the user's high priorities and the preferred episodic reward, which can be trivially observed from the directional derivatives of the scalarisation function. Combining this scalarisation function with the Approximator's estimate of each state-policy pair's episodic rewards, an estimate of the user's utility for each state-policy pair can be calculated, and the policy with the maximal predicted utility can be performed in a given state. While predicting the outcome of complete policies is not trivial, it has been successfully done by Adaptive Agent Team (2023), who used transformers to predict the environment's states and resulting rewards four actions ahead with reasonable accuracy. If there is inherent uncertainty present due to stochastic transitions, the Approximator's policy can be shortened to a suitable length, such as one action.

Summarizing, a diagram of the proposed solution is presented in Figure 2, while the pseudo-code for the proposed solution is presented as Algorithm 1. We claim the scalarisation function approximation presents an effective and straightforward opportunity to compute an user-preferred solution to an MOMDP under the ESR criterion, and to prove this opportunity pays well too, we have implemented the Conciliator steering and designed an experiment for it, detailed in the following section.

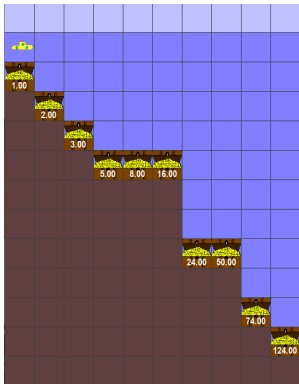

(a) A screenshot of the DeepSeaTreasure v1 environment with the recommended standard treasure chests and the tile grid shown. An agent is controlling a submarine that needs to navigate to one of the treasure chests in the seabed while optimising time and fuel used during the trip and the amount of treasure discovered.

(b) A table describing the locations and values of the chests in the recommended standard setting of the DeepSeaTreasure v1.

| Location | Value |
|----------|-------|
| (0, 1)   | 1.0   |
| (1, 2)   | 2.0   |
| (2, 3)   | 3.0   |
| (3, 4)   | 5.0   |
| (4, 4)   | 8.0   |
| (5, 4)   | 16.0  |
| (6, 7)   | 24.0  |
| (7, 7)   | 50.0  |
| (8, 9)   | 74.0  |
| (9, 10)  | 124.0 |

Figure 3: A screenshot and details from the DeepSeaTreasure v1 environment.

## 5 Experiments

There is a lack of modern and complex standardised benchmarks for discrete MORL decision-making problems Cassimon et al. (2022). Many of the state-of-the-art RL benchmarks, such as the Atari games or the Mountain Car problem, feature the problematic assumption of a single optimal policy, rendering the multi-policy approaches unnecessary. Additionally, the available benchmarks do not feature a comparable complexity in the size of action and state spaces, and many of them have not been adopted into wide use. The other aspect that should be considered here is the need of discrete state and action spaces for Conciliator steering.

Consquently, there is only one viable test environment left for the Conciliator steering: the DeepSeaTreasure v1 proposed by Cassimon et al. (2022). The DeepSeaTreasure introduces a simple, configurable and computationally lightweight decision-making problem with three objectives and a known Pareto front. This motivation quite neatly answers for the need of Conciliator steering to be applied in a decison-making MORL problem formulated as an MOMDP problem with ESR criterion. As there are no other test environments available that would provide the needed benchmark for our work and modifying the existing ones would be a research topic of its own, we need to leave the development of additional benchmarks for future research.

In the standard setting of the benchmark, a submarine is expected to navigate to one of 10 different treasure chests lying at the bottom of the seabed under a given time limit while optimizing the amount of discovered treasure, consumed fuel and spent time during the trip. As an action space, the submarine has 49 different two-dimensional accelerations to choose from, split into horizontal and vertical axes with negative and positive directions (the acceleration is capped at 3 units per direction). The accelerations are then mapped into $(2n + 1)^2$ different velocity vectors determining the submarine's movements in a grid world, where the $n$ is a user-assigned maximum velocity per direction. In the recommended standard setting by Cassimon et al. $n = 5$, but in our tests, we chose to use $n = 4$. We will elaborate this choice further on Section 7. The treasure chests' locations and values presented in Table 3b, whereas the initial position of the submarine is $(0, 0)$. Additionally, the environment can be configured with a more complex seabed, but as Cassimon et al. (2022) pointed out, the standard seabed makes the results easier to compare across studies, we shall report our results using the standard seabed.

To complicate the problem, the submarine can "coast", ie. not accelerate at all, preserving its gained speed and saving fuel the acceleration would consume. Additionally, the submarine can collide into the edges or the seabed, which would nullify the gained speed but not move the submarine. The state of the submarine consists of the submarine's current speed and its relative distances to each treasure chest as a Manhattan metric in $x$- and $y$-directions. The agent's goal is to choose a chest according to the preferred trade-offs

between the objectives and formulate the best route to the chest as a policy, consisting of a list of acceleration tuples. An illustration of the simulation environment is shown in Figure 3a, with the tile grid rendered to better show how the submarine can move in the environment.

While the DeepSeaTreasure v1 is indeed a good fit for the Conciliator steering in general, it is also notably idealistic in comparison to the real-world decision-making problems. The environment model is wholly deterministic and has an analytical formulation, which can be easily given to the agent. Given that and the relatively small and discrete state-action space, there is no point in testing different environment models for the role of the Approximator while using this specific benchmark. Thus the testing of more complex Approximators in stochastic environments is left for future work, and viable alternatives for environment models are presented in Section 2. The Approximator in the implementation can be replaced by any model capable of estimating the resulting rewards for a given policy. To keep this estimate exact, we chose $\gamma = 1$ and set the length of Approximator's policies in one action for our experiments.

Concluding from this, the benchmark is best suited to testing explicitly the Conciliator and how well it finds the routes for different preferred outcomes. As such, the experiment is designed with this explicit purpose in mind: the experiment shows whether the Conciliator steering is capable of handling different user profiles and manages to find a satisfactory policy for the user. Consequently, the traditional notion of different solution sets, such as Pareto fronts and coverage sets, is not applicable here. If the user should prefer a reward close to the Pareto optimal reward and the priority weighting would match the rewards gained by that weighting, then there is a theoretical possibility but not a guarantee that Conciliator steering returns this exact policy. Thus we do not report our results using traditional metrics, such as the sparsity or hypervolume of the approximated Pareto front, but resort to comparing the difference in rewards resulting from the policy selected by the Conciliator with respect to the reward preferred by the user as well as the maximal step-wise similarity of the selected policy in respect to another Pareto optimal policy as well as their component-wise difference. These metrics will reveal whether the Conciliator steering is capable of satisfying the user's preferences and yet differentiate the possible sub-optimality of the user's preferred reward with respect to the Pareto optimality.

Continuing from the definition of the Conciliator, it requires no training, but only the reward vector $\mathbf{r}$ and possible transformations of this vector, denoted by $f, f \colon \mathbb{R}^3 \to \mathbb{R}^3$, and the priority weighting of the user to function. Here we used the following function for the transformation:

$$f(\mathbf{r}_i) = \begin{cases} \mathbf{r}_i/60, \mathbf{r}_i > 0 \\ e^{\mathbf{r}_i/10}, \mathbf{r}_i \leq 0. \end{cases} \tag{3}$$

This function was chosen due to the properties of exponent function — it maps negative rewards into positive ones — and the scaling factor, which maps the treasure component into a roughly same magnitude as the other transformed components.

The Approximator in turn was pre-defined following the exact analytical environment model and reward functions for one time step, requiring no training either. The exact formulations for the reward estimates in a given time step $t$ are the following equations:

$$time(t+1) = \begin{cases} -t-1, \text{ if } a_t \text{ legal} \\ -\infty, \text{ if } a_t \text{ illegal} \end{cases}$$

$$fuel(t+1) = \begin{cases} fuel(t) - ||a_t||^2, \text{ if } a_t \text{ legal} \\ -\infty, \text{ if } a_t \text{ illegal} \end{cases}$$

$$treasure(t+1) = \begin{cases} \sum_{k=1}^{10} treasure_k \cdot w_k, \text{ if } a_t \text{ legal} \\ -\infty, \text{ if } a_t \text{ illegal}, \end{cases}$$

$$w_k = e^{-d((x_t,y_t),(x_k,y_k))}, k \in \{1, 2, ..., 10\},$$

$$d((x_t,y_t),(x,y)) = |x - x_t| + |y - y_t|.$$

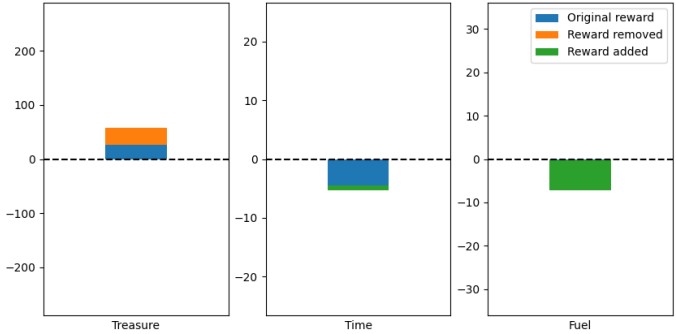

(a) Priority weighting of $\mathbf{p} = (1/10, 2/10, 7/10)$ and a preferred reward of $\mathbf{r}' = (25.81, -4.56, -0.27)$.

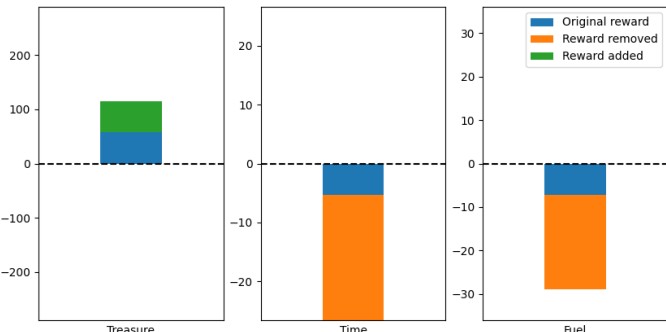

(b) Priority weighting of $\mathbf{p} = (98/100, 1/100, 1/100)$ and a preferred reward of $\mathbf{r}' = (115.6, -28.93, -28.93)$.

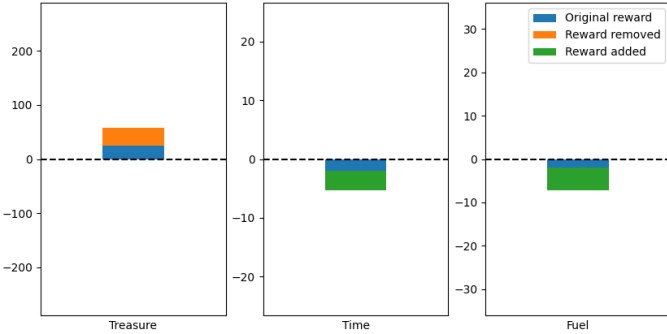

(c) Priority weighting of $\mathbf{p} = (1/5, 2/5, 2/5)$ and a preferred reward of $\mathbf{r}' = (24.45, -2.05, -2.05)$.

Figure 4: The plots of transfers under different user profiles. The baseline reward $\mathbf{r}$ for all the transfers was $(57.80, -5.32, -7.20)$.

The weighting function is designed to mitigate the influence of far-away high rewards while keeping near low rewards important. The treasures are still filtered according to the user's preference before the weighting, by eliminating the lower treasures out of the weighting.

Table 2: The specifications of the equipment used in the experiment as well as the resulting power and energy consumption and carbon emissions on average over the experiment, reported up to three decimals' accuracy.

| OS | Python version | CodeCarbon version |
|---|---|---|
| Windows 10-10.0 | 3.9.2 | 2.3.2 |

| CPU model | CPU count | RAM size in GB |
|---|---|---|
| Intel(R) Core(TM) i5-9300H CPU @ 2.40GHz | 8 | 16 |

| GPU model | GPU count |
|---|---|
| GeForce GTX 1650 with Max-Q Design | 1 |

| CPU power | GPU power | RAM power |
|---|---|---|
| 42.5 W | 3.634003 W | 5.942914 W |

| Average energy used per CPU | Average energy used per GPU | Average energy used per RAM |
|---|---|---|
| 0.257 Wh | 0.0213 Wh | 0.0000388 Wh |

| Average duration | Average energy used | Average emissions rate | Average emissions |
|---|---|---|---|
| 21.770 seconds | 0.279 Wh | 0.00186 g/s | 0.0406 g of $CO_2$eq |

After this initial programming of the Approximator, the Conciliator was tasked right away with finding one policy that produce rewards best matching the user's preference. The time limit for the maximum duration of the policy was 50 time steps. As the baseline reward $\mathbf{r}$, we used the vector $(57.80, -5.32, -7.20)$, determined as the average of the resulting episodic rewards from the Pareto optimal policies recorded in the dataset. Three different user profiles were used: first, the priority weighting of $\mathbf{p} = (1/10, 2/10, 7/10)$ and the preferred reward of $\mathbf{r}' = (25.81, -4.56, -0.27)$; second, the priority weighting of $\mathbf{p} = (98/100, 1/100, 1/100)$ and the preferred reward of $\mathbf{r}' = (115.6, -28.93, -28.93)$; and third, the priority weighting of $\mathbf{p} = (1/5, 2/5, 2/5)$ and the preferred reward of $\mathbf{r}' = (24.45, -2.05, -2.05)$. These user profiles are illustrated in Figure 4. After specifying the user profiles, the Conciliator steering sought out one policy for each user profile in both cases under ESR criterion, choosing the action that was estimated to produce maximal return for the user. The code for the experiment and the algorithm's implementation is available in GitHub. A modified version of DeepSeaTreasure library is also included, as our implementation introduces several bug fixes that are not available via PyPI DeepSeaTreasure v1 at the time of writing.

## 6 Results

The constructed policies are reported in Table 3, along with their similarity to a solution belonging to the Pareto front. The solutions from the front were chosen as the ones most closest to the policy constructed by the Conciliator steering, measured by the episodic rewards' component-wise difference and the actions taken in each time step.

As we can note, the Conciliator steering's selected policies are identical to Pareto optimal policies in the first and last case. In the second case, the selected and Pareto optimal policy differ significantly, while their rewards differ only by the fuel component. We can also note that the preferred and received rewards differ in each profile, but the differences can be rather small in one component that has higher priority than other components. Thus the Conciliator steering has selected the higher priority objective's performance over lower priority objectives, as indeed it should. Likewise, some of the differences actually make the solution better in a Pareto optimal sense by using less fuel or time than the user preferred, which is still considerably far away from the Pareto optimal policy's reward. Summarizing, we can note that Conciliator steering produced satisfactory policies for the user with negligible bias in the sense that the selected policies' trade-offs follow

Table 3: The policies discovered under each user profile and their corresponding rewards, as well as the preferred rewards and a few selected Pareto optimal policies and their rewards.

| | |
|---|---|
| Priority weighting $\mathbf{p}$ | $(1/10, 2/10, 7/10)$ |
| Preferred reward $\mathbf{r}'$ | $(25.81, -4.56, -0.27)$ |
| Received reward (treasure, time, fuel) $\mathbf{r}_\pi$ | $(0, -50, 0)$ |
| Pareto optimal reward (treasure, time, fuel) $\mathbf{r}^*$ | $(0, -50, 0)$ |
| Component-wise difference between a Pareto optimal $\mathbf{r}^*$ and received reward (treasure, time, fuel) $\mathbf{r}_\pi$ | $(0, 0, 0)$ |
| Component-wise difference between received $\mathbf{r}_\pi$ and preferred reward $\mathbf{r}'$ (treasure, time, fuel) | $(-25.81, -45.44, 0.27)$ |
| Component-wise difference between a Pareto optimal $\mathbf{r}^*$ and preferred reward $\mathbf{r}'$ (treasure, time, fuel) | $(-25.81, -45.44, 0.27)$ |
| Selected policy $\pi$ | Idle at initial position until time limit |
| Pareto optimal policy $\pi^*$ | Idle at initial position until time limit |

| | |
|---|---|
| Priority weighting $\mathbf{p}$ | $(98/100, 1/100, 1/100)$ |
| Preferred reward $\mathbf{r}'$ | $(115.6, -28.93, -28.93)$ |
| Received reward (treasure, time, fuel) $\mathbf{r}_\pi$ | $(124, -4, -44)$ |
| Pareto optimal reward (treasure, time, fuel) $\mathbf{r}^*$ | $(124, -4, -22)$ |
| Component-wise difference between a Pareto optimal $\mathbf{r}^*$ and received reward (treasure, time, fuel) $\mathbf{r}_\pi$ | $(0, 0, -22)$ |
| Component-wise difference between received $\mathbf{r}_\pi$ and preferred reward $\mathbf{r}'$ (treasure, time, fuel) | $(8.4, -24.93, -15.07)$ |
| Component-wise difference between a Pareto optimal $\mathbf{r}^*$ and preferred reward $\mathbf{r}'$ (treasure, time, fuel) | $(8.4, -24.93, 6.93)$ |
| Selected policy $\pi$ | $[[3, 3], [1, -2], [-2, 3], [-2, -2]]$ |
| Pareto optimal policy $\pi^*$ | $[[2, 1], [2, 1], [-1, 1], [-3, 1]]$ |

| | |
|---|---|
| Priority weighting $\mathbf{p}$ | $(1/5, 2/5, 2/5)$ |
| Preferred reward $\mathbf{r}'$ | $(24.45, -2.05, -2.05)$ |
| Received reward (treasure, time, fuel) $\mathbf{r}_\pi$ | $(8, -4, -2)$ |
| Pareto optimal reward (treasure, time, fuel) $\mathbf{r}^*$ | $(8, -4, -2)$ |
| Component-wise difference between Pareto optimal and received reward (treasure, time, fuel) $\mathbf{r}_\pi$ | $(0, 0, 0)$ |
| Component-wise difference between received $\mathbf{r}_\pi$ and preferred reward $\mathbf{r}'$ (treasure, time, fuel) | $(-16.45, -1.95, 0.05)$ |
| Component-wise difference between Pareto optimal and preferred reward $\mathbf{r}'$ (treasure, time, fuel) | $(-16.45, -1.95, 0.05)$ |
| Selected policy $\pi$ | $[[1, 1], [0, 0], [0, 0], [0, 0]]$ |
| Pareto optimal policy $\pi^*$ | $[[1, 1], [0, 0], [0, 0], [0, 0]]$ |

the priority weighting the user assigned and the received reward is close to or better than the preferred reward, even when the preferred reward is not close to the Pareto optimal policy's resulting reward.

Additionally, to showcase the light computational burden of the Conciliator steering, the power consumption and carbon emissions estimated with CodeCarbon's process tracking mode as an average over the experiment runs in the experiment are reported in Table 2, along with the technical specifications of laptop that was used for the experiment. When using the same tool, the emissions and energy consumption of a fully connected MNIST classifier's training on a personal laptop were reported to be 0.056 g/$CO_2$eq and 1 Wh by Bouza

et al. (2023). In comparison, we can conclude the Conciliator steering's emissions and energy consumption are roughly a quarter and three quarters of the MNIST classifier's respectively, and thus minimal.

## 7 Discussion

A notable hindrance of the Conciliator steering lies in the assumption of positive rewards. Most real-world problems include sanctions, so it would be beneficial to extend the reward transfer to apply to negative rewards as well. However, the transfer optimization for negative rewards is not mathematically trivial, as the bounds of the transfer are not symmetrical across the reward components. The negative rewards, along with possibly varying magnitude of positive rewards, also complicate the choice of a possible mapping of the transfer back to the preferred reward in a scale that matches the problem's definition of the reward function. In practice, if the magnitude of reward components varies, the component with the largest magnitude affects the transfer most even with small changes of the priority weighting. As done here, such behaviour can be mitigated by a suitable mapping of such rewards for the transfer calculation. For example, in the DeepSeaTreasure v1 without a mapping, the treasure is the reward component with the highest magnitude in most cases, with a low priority for treasure and slightly higher priorities for fuel and time, all reward components would end up close to zero, which is not realistic.

In future however, the applications could be extended even more if such mappings would not be required or the optimization algorithm could adapt to the varying scale of the rewards, as the mappings can not generally be transferable between different reward functions. Another approach to mitigate this behaviour would to scale the transfer so that it is not one-to-one between the rewards, which then could be adapted across tasks according to the reward function.

Additionally, it can be noted that the convergence of the Conciliator steering to the user-preferred policy is highly dependent on the complexity of the user's priority weighting. While there is a guaranteed global minimum given the priority weighting and the SHGO is guaranteed to find this minimum Endres et al. (2018), very little can be said how fast the user finds the correct priority weighting for the preferred reward. However, as the user can freely iterate the priority weighting with a an arbitrarily fine-grained slider before moving on to the policy selection, this practical problem does not pose a significant computational hindrance. The convergence can be hastened by having a realistic initial baseline for the reward, reflecting the true total reward attained on average without any weighting. This baseline will most likely result in functional weights for the priority weighting and consequently the scalarisation function, which is why the average reward over Pareto optimal policies' rewards is chosen here as the initial reward.

A weak link in the Conciliator steering is the accuracy and foresight of the Approximator. In our tests, we discovered that using $n = 5$ as the maximum velocity limit leads the Approximator to the edge in a maximum velocity. There the submarine cannot move anymore as the preferred fuel is consumed already. This dead end is due to the fact that the Approximator can only see one time step ahead and the maximum velocity of 5 can only be gained and reversed in four time steps in total. Thus choosing a more conservative limit for maximum velocity of 4, resulting in less fuel consumption, the Approximator's foresight is long and accurate enough for the problem. Thus one advancement could be directed towards more complex Approximators with longer foresights and an ability to handle stochastic transitions. In this case, it would be beneficial to extend the Conciliator steering so that it can discover multiple policies in one run.

Another advancement can be directed to the exact formulation of the scalarisation function with the proposed weights. The difficulty of this avenue lies in the justification of the choice: the problem formulation assumes the scalarisation function to be unknown by the user, and therefore any distance function to the preferred reward vector or a monotonically increasing can be equally apt, as proven by Wierzbicki (1982). While the choice of $L_p$-norms an approximation is reasonable given the reference point technique, ie. the idea that the user's preference encapsulates the scalarisation function, many other functions could be fitted still, if given suitable conditions in the definition.

Finally the practical applicability of the Conciliator is hindered by the dimensionality of the optimization needed in the transfer vector, as SHGO can handle roughly 10 dimensions with reasonable computation time Endres et al. (2018). By introducing more constraints for the optimization problem, there is a possibility

another optimization algorithm could be extended into more dimensions in reasonable time. However, the applicability of such constraints can be specific to the use case, so they shall not be addressed here.

All in all, the Conciliator's applications are still wide in decision-making problems, where the ESR condition is present. Additionally, our intended next step for future research is to extend the Conciliator steering to the SER condition and test it with partially observable MOMDPs and more complex environments with continuous states. Extension to SER would require the Conciliator to account for the expected rewards of different outcomes of the same policy, as illustrated in Table 1. As the calculation is now conducted using episodic rewards and the scalarisation function is directly connected to the distance between the resulting and preferred reward, the calculations would have to be changed to regard expected episodic rewards instead of episodic rewards themselves. The extension to partially observable MOMDPs and more complex environments with continuous states would require different Approximators.

The very first application where we will research these limitations is the electric vehicle charging station placement problem, where the agent's goal is to place a pre-defined amount of charging stations in a fixed road network while optimizing the outcome according to the user's preferences between the objectives of resulting carbon dioxide emissions, the average queuing time across the stations and maximal steady traffic flow in the central roads over the time span of one day. This problem requires more sophisticated Approximator architecture, which could for example be based on the transformer architecture of AdA by Adaptive Agent Team (2023), or the RNN-LSTM by Ha & Schmidhuber (2018), as there is no analytical model for the emission and traffic flow estimation available. The used optimality criterion is still ESR, as the whole network of stations and their locations is modelled as one action, and the station network is thus built only once.

After the station placement we will focus on simulating the consumer demand for the new charging stations, where the agent is the driver optimizing the distance to the station, the queuing time, the price and the functionality of the station, while having a fixed need of charge during the episode of one week. The action space consists of the routing made by the agent during their trip. As the agent is expected to repeat the chosen policy more than once during the week, the optimality criterion is SER. Likewise, as the agent has no access to the info regarding all the charging stations in the network, the problem is partially observable. The state of charge is also a continuously changing variable, thus representing the need for Approximator to deal with continuously distributed states.

## 7.1 Statement of Broader Impact

Summarizing discussion in this statement, the Conciliator steering can be used in decision-making problems which are difficult to optimize due to conflicting objectives. As these problems holds considerable research potential and public benefit, also the Conciliator steering can be deemed to have significant potential and benefit for its ability to solve them in a simple and user-preferred manner. The second factor in Conciliator steering's broader impact can made from the carbon footprint caused by the development of AI, as noted in Bouza et al. (2023). The Conciliator steering is especially designed to be a lightweight algorithm without the need for training, and this is showcased in the estimates produced by CodeCarbon Python package developed by Courty & Schmidt (2023). As such, it can be said that Conciliator steering is an eco-friendly AI that still produces good results. We hope that this reporting will encourage other researchers to develop eco-friendly AIs and publish their emission and power consumption reports as well. Furthermore, the paper presents extensively the discussion and related works related to human-centered RL, highlighting the emerging problems and possible avenues at length to pave way for new scientific breakthroughs. Finally, we note that there is a chance of using the Conciliator steering to optimize for ethically questionable objectives, such as solely for attaining financial gain in the traffic organization. However, limiting different objectives' optimization cannot realistically be accounted for in the algorithm development due to complex contexts present. Consequently, we remind users to apply Conciliator steering responsibly and proceed with caution and in accordance with relevant and applicable laws when implementing any decisions in the real world based on the results of the algorithm.

## 8   Conclusions

Much research has been devoted to RL, but less attention in the field has been given to optimising MORL problems, especially those with user preferences and without linear scalarisation. One of the challenges in this problem lies in finding suitable optimization methods, another in the approximation of the user's scalarisation function, and finally in the different optimality criteria rising in real-world problems, SER and ESR. This paper addressed these issues by presenting a novel algorithm called Conciliator steering that uses the concepts of reward transfer and priority weighting to approximate user-preferred scalarisation function and policies in a MOMDP framework under ESR criterion, which has not been used before for human-centered RL. Via a simple and effortless interaction between the user and the agent requiring no pre-defined user information, we proved that Conciliator steering produces satisfactory results in DeepSeaTreasure v1 environment with only negligible bias while being computationally lightweight, which has not been achieved before.

## Acknowledgements

This work was supported by the Research Council of Finland project 347197 Artificial Intelligence for Urban Low-Emission Autonomous Traffic (AIforLEssAuto), the Research Council of Finland Flagship program: Finnish Center for Artificial Intelligence (FCAI) and the the Department of Computer Science, University of Helsinki. This work was supported by the Research Council of Finland (RCF) project 347197 Artificial Intelligence for Urban Low-Emission Autonomous Traffic (AIforLEssAuto), RCF project 332177 Sustainable urban development emerging from the merger of cutting-edge Climate, Social and Computer Sciences (CouSCOUs), RCF Flagship program: Finnish Center for Artificial Intelligence (FCAI), and the Department of Computer Science, University of Helsinki.

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
