# OpenReview forum: "Conciliator steering: Imposing user preference in multi-objective reinforcement learning"
_TMLR — Accepted by TMLR_

### Review · Reviewer_H6QA · 2024-02-07

**Summary Of Contributions:**

The paper presents "Conciliator steering," a novel algorithm designed for multi-objective reinforcement learning (MORL) that aims to address the challenge of adapting to user preferences in the presence of multiple objectives. It leverages priority order and reward transfer to approximate the scalarisation function, thereby producing policies that align with user preferences under the expected scalarised returns (ESR) criterion. The algorithm was tested on the DeepSeaTreasure v1 benchmark, demonstrating its ability to generate user-preferred policies with minimal user-agent interaction and computational overhead. The study emphasizes the efficiency of Conciliator steering in terms of both computational resources and environmental impact, highlighting its low carbon dioxide emissions and energy consumption compared to traditional methods.

**Audience:**

Yes

**Claims And Evidence:**

No

**Requested Changes:**

I would like to see more extensive experiments on multiple domains with many MORL baselines. Additionally, I would like more extended discussion between the similarity and difference between the approximator and value function. So I would like these additional experiments and discussion to believe that the claims are sufficiently supported.

**Strengths And Weaknesses:**

Strengths:
- User-Centered Design: The algorithm focuses on simplifying the interaction between the user and the agent, making it easier for users to provide constructive feedback.

- Computational Efficiency: Conciliator steering is shown to be computationally lightweight, producing significantly lower carbon dioxide emissions and energy consumption compared to a fully connected MNIST classifier training.

Weaknesses:

- Limited Testing Environment: The algorithm was only tested on the DeepSeaTreasure v1 benchmark. Additional studies on diverse and complex environments are necessary to validate its effectiveness and generalizability.
- Comparison with Other Approaches: The paper lacks a detailed comparison with other existing MORL solutions, which would help in understanding the relative advantages and improvements offered by Conciliator steering.

---

### Review · Reviewer_A1ex · 2024-02-19

**Summary Of Contributions:**

This paper deals with generating policies according to user preferences over a multi-objective sequential decision making task. It does so by combining two components: a model-based method that can predict the performance of a given policy (called the approximator), and a policy selection method based on user-preferences (called the conciliator).

The user preferences are communicated via a priority ordering on the different objectives, as well as a preferred return value.
The proposed solution then finds a policy with returns as close to the preferred returns with prioritization done via the priority ordering.
Experiments are conducted on the DeepSeaTreasure benchmark environment showing that pareto-optimal policies are found if the approximator is accurate enough and the priority ordering and preferred returns are not explicitly optimizing for policies not on the front. When the policy that best captures user preferences is not on the front, a policy that is close to the users preferences as well as the pareto front is returned.

**Audience:**

Yes

**Broader Impact Concerns:**

there is a broader impact statement included. The term `the final responsibility and consequences of using Conciliator steering for such purposes rests with the user and not with the AI developer` might need to be expanded and dealt with in more nuance.

**Claims And Evidence:**

No

**Requested Changes:**

Most of the weaknesses above have been written in a way that they can be considered requested changes.

**Strengths And Weaknesses:**

## Strengths:
* This paper takes an interesting view of solving multi-objective tasks, by breaking them down into the approximator and the conciliator.
* The method of finding the preferred policy seems general and useful from a high level.
* Experiments seem to validate the claims made in the paper.

## Weaknesses:
Section 4, the proposed solution, is not written clearly enough to judge the correctness of the proposed technique. It also does not go into enough detail and does not provide rigorous proofs of any claims made there.
I will focus the next bullet points on places where this section can and should be improved.
* In Definition 3, it is unclear if r and r' individual rewards, specific functions, or the episodic returns? It is unclear even from context and from the experiments.
* Definition 4, the priority ordering is a misnomer. This vector is not giving an ordering, but a weighting. The weighting can be used to deduce an ordering.
* Definition 5: since the definition of r and r' has become confusing, it is unclear what t is.
* Statement `... the lower priority objectives will start to transfer their reward to the higher priority objectives and vice versa ...` in Section 4 (page 8) is hard to follow.
* Similar to the above, in the sentence ending with `... this results in preferred rewards r′ that are bounded in distance with respect to the reward r.` the first two statements in the sentence are fine. However, how they imply this conclusion is not clear. The authors should expand on it or prove this conclusion.
* It is never explicitly stated what parameters are being optimized in Equation 1.
* The constraint $\sum_{i=1}^{n}p_i = n$ has not been discussed in the definition or clarified before or after Equation 2.
* On page 8, paragraph under Equation 2, there is a sentence ending with `... one reward can not gain infinitely by .` This sentence is incomplete.
* The positive Hessian of the transfer vector is specified in Equation 3 without any additional explanation. This claim needs some proof.
* On page 7, the statement `As the formulation of different rewards is often fairly straightforward and intuitive for users, ...` needs to be backed up by references or proof. There have been numerous works [1, 2, 3] which show that humans are not good at designing reward functions.

The rest of this section, I will go over some of the other sections:
* The introduction dives into details without giving a general overview. Less technical details and more general overview would be good to have here.
* The introduction gives three possible ways in which the scalarisation function can be approximated. Illustrations of how these three methods could be used in an example domain would be useful and easier for a reader to digest.
* In Section 2.1, a Pareto front is mentioned without definition or reference.
* Last paragraph of page 2, the statement `The primary solution presented so far in the literature is often referred to as reward shaping or
reward engineering, i.e. the careful hand-crafting of the reward and scalarisation functions to match the scales of returns to the human’s preference.` needs some reference.
* Page 4, `A recent treatise into the topic of human-aligned RL has been written by Sutton & Barto (2018)`. Sutton and Barto's book does not focus on human-in-the-loop or human aligned RL. That reference would be more appropriate as an overview of RL.
* There are multiple references such as Hayes et al. In Section 2.1, first paragraph, third from the last line; Vamplew et al. on page 4 midway through the last paragraph; Cassimon et al.  on page 13; where the reference is incomplete.
* In Section 3, the reward is defined as a $d$ dimensional vector $(\mathtt{R}: S \times A \times S \rightarrow \mathbb{R}^d)$, but then the text says `... with one component for each of the n objectives.` Stick to $d$ or $n$ for the number of objectives.
* For the definition of the value function, Is the expectation is over just time-steps, or over the time-steps and the $d$ dimensional reward function as well?
* The paper goes over the expected scalarised returns (ESR) and scalarised expected returns (SER) criterion in great detail, but does not use this information. Perhaps this distinction can be introduced briefly instead of spending so much space on a difference that does not matter in the main method of the paper.

---

### Review · Reviewer_SKuq · 2024-03-11

**Summary Of Contributions:**

This paper studies multi-objective reinforcement learning (MORL) problems, and as a solution, proposes an algorithm called "Conciliator steering" (Algorithm 1). Preliminary experiment results are also demonstrated (Section 5, 6).

**Audience:**

Yes

**Broader Impact Concerns:**

My understanding is that the broader impact concerns have been sufficiently addressed in the "Statement of Broader Impact" section (Section 7.1).

**Claims And Evidence:**

Yes

**Requested Changes:**

- Please further improve the writing of this paper.

- Please provide more justification for the proposed "Conciliator steering" algorithm. It can be a theoretical analysis, or more experiment results, or both.

**Strengths And Weaknesses:**

Strengths:

- This paper has done a good job of literature review.

Weaknesses:

- Overall, I feel that the core ideas behind the proposed "Conciliator steering" algorithm are quite straightforward. My impression is that this paper lacks the sophistication and novelty that a paper published in a top-tier machine learning conference/journal should have.

- The writing of this paper can be further improved. In particular, I feel that Section 4 of this paper is too lengthy and can be significantly shortened.

- I also feel that we need more justification for the proposed "Conciliator steering" algorithm. It can be either a theoretical analysis, or more experiment results.

---

> ### Author Response · Authors · 2024-03-22
> **Thank you for the suggestions**
>
> We thank you for your valuable feedback. We have now uploaded a revision of the manuscript, which we refer to as the final version from now on.
>
> In the final version, we performed the following changes:
>
> 1. We discussed more about the transfer capabilities of Approximator in the discussion section, as requested in the first review.
> 2. The problem formulation, discussion, proposed solution and results have been clarified mathematically and the term priority order changed to priority weighting to better address the mathematical concerns raised in the first and second review. The difference of value function, the ambiguous parts in mathematics of the proposed solution and relevancy between SER and ESR regarding the proposed solution are now all addressed.
> 3. We improved the writing of introduction and related work as requested in the second review, following closely our reply to the second review. The changes at large were restructuring sentences, adding references and removing repetitive or irrelevant texts in the related work while re-writing the introduction again from the basis of practical examples as suggested in the second review.
> 4. We improved the writing regarding the section of proposed solution, as requested in the third review. We did it by restructuring and removing content from the general outline at the start of the section while condensing the discussion relating to the suboptimality of human-centered AI into its own paragraph later in the section. Due to these changes requested and the mathematical concerns raised, the section's length regarding the proposed solution did not shorten as requested in the third review; instead, it increased slightly. We did keep this point in mind when revising the writing and strived to justify all the presented content and its relevancy.
> 5. As we replied to the first review, adding more experiments in other domains is not possible without further research and more experiments in the same environment will not add more justification for the algorithm, as it will not adapt in anyway to a new configuration in the same environment. A more detailed mathematical analysis of the algorithm can be provided only on a case-by-case basis, and as such, cannot be added to the manuscript, which is primarily addressing the general outline. The experiment section in turn addresses the specific concerns of that case. Consequently, no further experiments were added in the final version.
> 6. More theoretical evidence about the algorithm in the form of an additional proof, middle calculations and more in-depth discussion about the different aspects and challenges were added into the section regarding the proposed solution to better address the concerns of validity of claims and proposed solution, as raised in the first, second and third review.
> 7. Additionally, we rectified an unit conversion mistake in the emission and energy comparison between the Conciliator steering and Bouza’s report in the results section and modified the remark in the abstract to reflect this. We added a minor change of term to Table 1 for consistency. We also rounded Table 3 and the related results in text to two decimals to improve readability and clarity.
>
> Finally, we like to thank all our reviewers and the editor for their valuable contributions for the improvement of the manuscript. We hope we have carefully and duly addressed all your concerns by doing our best according to your feedback.

---

### Decision · Action_Editor_nqTW · 2024-05-17

**Recommendation:** Accept with minor revision

**Comment:**

The submission proposes an algorithm called “Conciliator steering” for multi-objective reinforcement learning (MORL) problems. The algorithm aims to generate policies that align with user preferences by leveraging priority order and reward transfer. However, the current submission has one weakness: Limited Experimental Validation. The algorithm was only tested on the DeepSeaTreasure v1 benchmark. More extensive experiments on multiple domains and comparisons with existing MORL solutions are recommended to validate its effectiveness and generalizability further.

Summary of Reviewers’ Feedback
- Reviewer SKuq: More justification for the algorithm is needed through theoretical analysis or additional experiments.
- Reviewer A1ex: The changes made in the revision are too numerous to evaluate comprehensively. The theoretical analysis needs to be more rigorous, and several definitions and claims are unclear or unsupported.
- Reviewer H6QA: While the paper presents an interesting approach, it can benefit from more extensive experiments and comparisons with other MORL solutions.

**Audience:**

Yes, the paper is of interest to TMLR’s audience. The focus on multi-objective reinforcement learning (MORL) and user-preference-based policy generation is relevant to the community. The paper contributes to the literature on adapting RL algorithms to user preferences, which is a significant area of interest.

**Claims And Evidence:**

The claims made in the submission are supported by accurate, convincing, and clear evidence.
The paper provides a solid framework for the proposed “Conciliator steering” algorithm.

Although the reviewers would love to see more theoretical support, such as a theoretical analysis and rigorous proof of the proposed technique, we believe the paper meets the TMLR standard, and the algorithm is interesting by itself to be published.

To further strengthen the paper, the author can consider providing additional justification for the algorithm through additional experiments, as suggested by the reviewers.

---

> ### Author Response · Authors · 2024-06-05
> **Thank you all for your final suggestions**
>
> Thank you all for your final suggestions. We addressed the concern of more extensive testing by explaining our approach for the tests in more detail at the beginning of the experiments section.